# Arginine: at the crossroads of nitrogen metabolism

Tak Shun Fung[1], Keun Woo Ryu[1] & Craig B Thompson [1,2]✉

## Abstract

L-arginine is the most nitrogen-rich amino acid, acting as a key precursor for the synthesis of nitrogen-containing metabolites and an essential intermediate in the clearance of excess nitrogen. Arginine's side chain possesses a guanidino group which has unique biochemical properties, and plays a primary role in nitrogen excretion (urea), cellular signaling (nitric oxide) and energy buffering (phosphocreatine). The post-translational modification of protein-incorporated arginine by guanidino-group methylation also contributes to epigenetic gene control. Most human cells do not synthesize sufficient arginine to meet demand and are dependent on exogenous arginine. Thus, dietary arginine plays an important role in maintaining health, particularly upon physiologic stress. How cells adapt to changes in extracellular arginine availability is unclear, mostly because nearly all tissue culture media are supplemented with supraphysiologic levels of arginine. Evidence is emerging that arginine-deficiency can influence disease progression. Here, we review new insights into the importance of arginine as a metabolite, emphasizing the central role of mitochondria in arginine synthesis/catabolism and the recent discovery that arginine can act as a signaling molecule regulating gene expression and organelle dynamics.

**Keywords** Arginine Metabolism; Mitochondria; Protein Synthesis; Metabolite Signaling; Arginine Deficiency
**Subject Categories** Cancer; Metabolism; Molecular Biology of Disease

## Introduction

Arginine is one of the 21 encoded amino acids incorporated into mammalian proteins (Chung and Krahn, 2022) and accounts for ~4.7% of the amino acids in proteins. Six mRNA codons (10%) code for arginine (Inouye et al, 2020), indicating its importance in contributing to protein structure and function. Arginine is classified as a semi-essential amino acid. Endogenous synthesis of arginine can be adequate to sustain adult health. Systemic levels of extracellular arginine are maintained by mitochondrial synthesis of citrulline in the small intestine from dietary glutamate and glutamine (Wijnands et al, 2015). Circulating citrulline is then converted to arginine in peripheral tissues, primarily the kidneys (Brosnan and Brosnan, 2004). However, during early development

or during infection, arginine synthesis alone is not sufficient to meet metabolic demands, and arginine uptake from exogenous sources becomes necessary (Morris, 2016).

Given the importance of arginine as both a protein component and a metabolic precursor, a holistic understanding of arginine production and utilization is needed. Here, we will explore the various facets of arginine biology that makes it a key component in protein synthesis, cellular metabolism, and intracellular signaling.

## Mitochondria play a central role in arginine metabolism

### Mitochondria are essential for de novo arginine synthesis

Mitochondria have diverse roles in cell biology but are best known for their role in energy production through oxidative phosphorylation (OXPHOS) (Picard et al, 2018). Less characterized is the central role mitochondria play in the incorporation of ammonia ($NH_4^+$) into organic precursors required for the synthesis of non/semi-essential amino acids including arginine as well as for the biosynthesis of polyamines and nucleotides (Kim et al, 2017) (Fig. 1). Two mitochondrial enzymes play roles in the incorporation of $NH_4^+$ into organic precursors for the de novo production of arginine. Glutamate dehydrogenase (GDH/GLUD1) catalyzes the production of glutamate from the tricarboxylic acid (TCA) cycle metabolite α-ketoglutarate (α-KG) and $NH_4^+$. Carbamoyl phosphate synthase 1 (CPS1) carries out the ATP-dependent condensation of bicarbonate ($HCO_3^-$) and $NH_4^+$ to form carbamoyl phosphate (CP). To produce arginine, mitochondrial ornithine transcarbamylase (OTC) catalyzes the production of citrulline from CP and ornithine. Citrulline exported out of mitochondria is then converted to arginine by the cytosolic enzymes in the urea/citrulline cycles. Depending on cellular conditions and/or cell type, the resulting arginine can be used to support protein production; used to clear excess $NH_4^+$ through the production of urea; used as a precursor in the production of the energy buffering phosphagen - creatine; used to synthesize agmatine and polyamines; or utilized as a signaling molecule to regulate lysosomal amino acid release into the cytosol (Fig. 2).

### Arginine can be produced from mitochondrial glutamine/glutamate

One major source of mitochondrial glutamate is through the deamination of glutamine. The glutamine concentration in the bloodstream is tightly maintained at around 500 μM (Zhang et al, 2017) making it the most abundant free extracellular amino acid (Hensley et al, 2013). By comparison, the physiological concentration of arginine in

[1]Cancer Biology and Genetics Program, Memorial Sloan Kettering Cancer Center, New York, NY 10065, USA. [2]Department of Medicine, Memorial Sloan Kettering Cancer Center, New York, NY 10065, USA. ✉E-mail: Thompsonc@mskcc.org

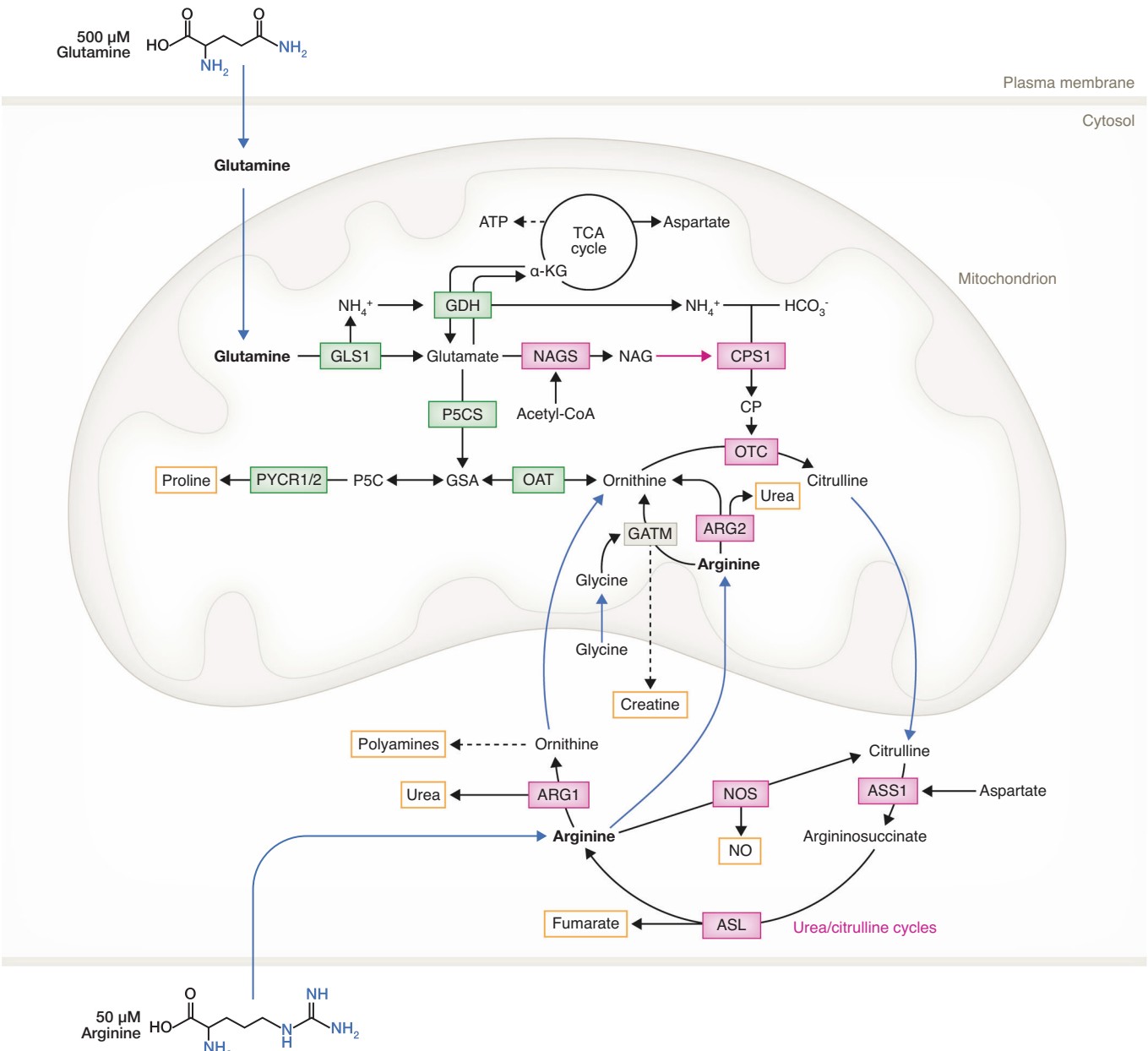

adult blood is 50 μM (Rodriguez et al, 2003). Extracellular glutamine is first taken up into cells through plasma membrane glutamine transporters and cytosolic glutamine is transported through the inner mitochondrial membrane (IMM) via a mitochondrial glutamine transporter SLC1A5 variant (Yoo et al, 2020). Once inside the mitochondrial matrix, glutamine is first deaminated by glutaminase 1 (GLS1) into glutamate and the reaction liberates $NH_4^+$ (Turner and McGivan, 2003). The $NH_4^+$ produced from the deamination of glutamine into glutamate can be utilized in CP synthesis (Kim et al, 2017) and/or used to convert α-KG to produce more glutamate (Spinelli et al, 2017). The nitrogen from two glutamates are used in the de novo mitochondrial synthesis of ornithine. The first glutamate can be processed into glutamate semialdehyde (GSA) by pyrroline-5-carboxylate synthase (P5CS) and the amino group from a second mitochondrial glutamate is transferred to GSA by ornithine

aminotransferase (OAT) resulting in ornithine. Therefore, mitochondrial $NH_4^+$ and glutamate—both contributed by glutamine—can be used in the synthesis of the arginine precursor ornithine. Mitochondrial ornithine is combined with CP by OTC to form citrulline and transported to the cytosol (Fig. 1).

## Citrulline-dependent production of arginine occurs in the cytosol

In the cytosol, argininosuccinate synthase (ASS1) condenses citrulline with aspartate to generate argininosuccinate (Apiz Saab et al, 2023; Khare et al, 2021; Kim et al, 2024). After which, argininosuccinate is broken down into arginine and fumarate by argininosuccinate lyase (ASL). The resulting fumarate can be taken back into the mitochondrial matrix and used to support TCA cycle anaplerosis.

**Figure 1. Glutamine and arginine metabolic pathways.**

Glutamine and arginine metabolism overlap in mitochondria. Once imported into mitochondria, glutamine is deaminated into glutamate by GLS1 which releases $NH_4^+$. Two pathways are present inside mitochondria to sequester $NH_4^+$: (1) α-KG accepts $NH_4^+$ to produce more glutamate catalyzed by GDH or (2) production of CP from $NH_4^+$ and $HCO_3^-$ catalyzed by CPS1. Once formed, CP can enter the urea cycle. Glutamate supports arginine synthesis in three different ways. Glutamate is a substrate, alongside acetyl-CoA, in the production of NAG which in turn activates CPS1. Glutamate can also be reduced to ornithine through sequential P5CS and OAT activity. Finally, glutamate is an anaplerotic substrate for the TCA cycle to produce aspartate. Both ornithine and aspartate are key components of the urea cycle. Ornithine can be converted to citrulline by addition of CP catalyzed by mitochondrial OTC. After which, citrulline is exported out into the cytosol for further processing by ASS1 and ASL to produce arginine. The reaction catalyzed by ASL releases fumarate, which can return to mitochondria as a TCA cycle intermediate. Arginine can be further hydrolyzed into ornithine by cytosolic ARG1 or the mitochondrial isoform, ARG2, releasing ornithine and urea as products. Mitochondrial arginine can also transfer its guanidino group to glycine catalyzed by GATM in the first biosynthetic step of the phosphagen creatine, while cytosolic arginine is used to produce NO. Chemical formula of arginine (bottom) and glutamine (upper) are shown. Each molecule of glutamine contains two reduced nitrogen (in blue) while one molecule of arginine contains four reduced nitrogen. Black arrows: single biochemical reaction. Blue arrows: translocation of metabolites across cellular compartments. Yellow boxes: Products of arginine metabolism. Magenta arrow: NAG is an allosteric cofactor of CPS1 for activation and not a substrate. Double arrows: reversible biochemical reaction. Dotted arrows: multi-step reaction to synthesize final product. Enzymes in the urea/citrulline cycles are in magenta while enzymes in the catabolism of glutamine/glutamate are in green. Creatine production enzyme GATM is in grey. The cyclization of GSA into P5C is a spontaneous reaction. NAGS N-acetylglutamate synthase, CPS1 carbamoyl phosphate synthetase 1, OTC ornithine transcarbamylase, ASS1 argininosuccinate synthase 1, ASL argininosuccinate lyase, ARG1 arginase 1, ARG2 arginase 2, GATM L-arginine:glycine amidinotransferase, NOS nitric oxide synthases, TCA cycle tricarboxylic acid cycle, GLS1 glutaminase 1, GDH glutamate dehydrogenase; P5CS pyrroline-5-carboxylate synthase; OAT ornithine aminotransferase; PYCR1/2 Pyrroline-5-carboxylate reductase 1 and 2; CP carbamoyl phosphate, NO nitric oxide, GSA glutamate γ-semialdehyde, P5C pyrroline-5-carboxylate, NAG N-acetylglutamate, α-KG α-ketoglutarate, Acetyl-CoA acetyl coenzyme A, ATP adenosine triphosphate.

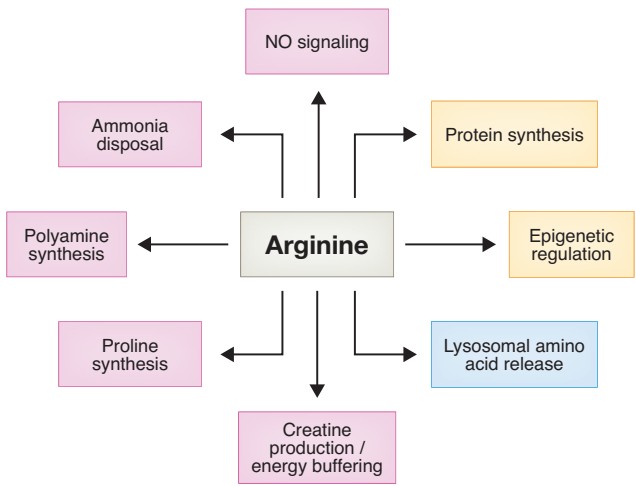

**Figure 2. The multiple roles of arginine in human health.**

In addition to protein synthesis, arginine is utilized as a signaling molecule, the precursor for a variety of important biomolecules and the protein residue for post-translational modifications.

The net production of arginine can be used to maintain transfer RNA (tRNA) charging and protein synthesis or converted into other metabolic intermediates that support cell bioenergetics, growth, or effector function.

In some cells, predominantly mature hepatocytes (Hajaj et al, 2024), macrophages (Yurdagul et al, 2020), and nonproliferating T cells (Tang et al, 2023) that either take up or produce arginine in excess of their needs, use arginine to eliminate excess nitrogen through the urea cycle. To produce urea, cytosolic arginine is hydrolyzed back into ornithine by either cytosolic arginase 1 (ARG1) or mitochondrial arginase 2 (ARG2) and urea is released. Urea is then transferred from the cell and excreted out of the body by the kidneys. Ornithine, on the other hand, is used by mitochondria to restart the cycle again or used in the cytosol as a precursor for the biosynthesis of other products such as polyamines (Hajaj et al, 2024).

Urea cycle enzymes are differentially expressed across cell types. Mitochondrial OTC and CPS1 are constitutively expressed mainly in hepatocytes and enterocytes (Marini, 2016). In contrast, cytosolic ASS1 is widely expressed across healthy tissues and can support de novo arginine synthesis (Ding et al, 2023). Renal cells express relatively high levels of both ASS1 and ASL but not CPS1 (Marini et al, 2017). This gives renal cells the ability to support the systemic arginine levels through the conversion of dietary citrulline to arginine (Brosnan and Brosnan, 2004).

Some cell types take up and/or synthesize arginine in support of nitric oxide (NO) synthesis (Fig. 1). In this cycle, arginine is converted to citrulline by nitric oxide synthases (NOS) (Förstermann and Sessa, 2012), producing NO. NO is an important regulator of vascular tone (Levine et al, 2012). Citrulline can be recycled back to arginine through ASS1 and ASL activity to complete this cycle. Most NOS isoforms exist within the cytosol but one (mtNOS) is found inside mitochondria (Ghafourifar and Cadenas, 2005). This suggests that arginine can enter mitochondria and raises the possibility that the activities of both ARG2 and mtNOS might regulate mitochondrial biology.

## Products of arginine and their roles

In addition to their use in protein synthesis, some amino acids are required precursors for the synthesis of other essential organic molecules. The use of glutamine (Zhang et al, 2017), serine, and glycine (Handzlik and Metallo, 2023) in the synthesis of other amino acids, lipids, and nucleotides is well-studied. In contrast, the role of arginine as a molecular precursor has been under-appreciated. Arginine is a highly versatile metabolite that serves as a precursor for multiple products (Fig. 3). However, unlike glutamine, serine, and glycine, arginine is semi-essential (Morris, 2016). To avoid this limitation, most modern tissue culture is supplemented with supraphysiologic levels of arginine (Salazar et al, 2016). How different cell types balance their use of arginine between protein synthesis and the support of these additional uses is poorly understood. In the following, we consider the role of arginine in the synthesis of other essential metabolites.

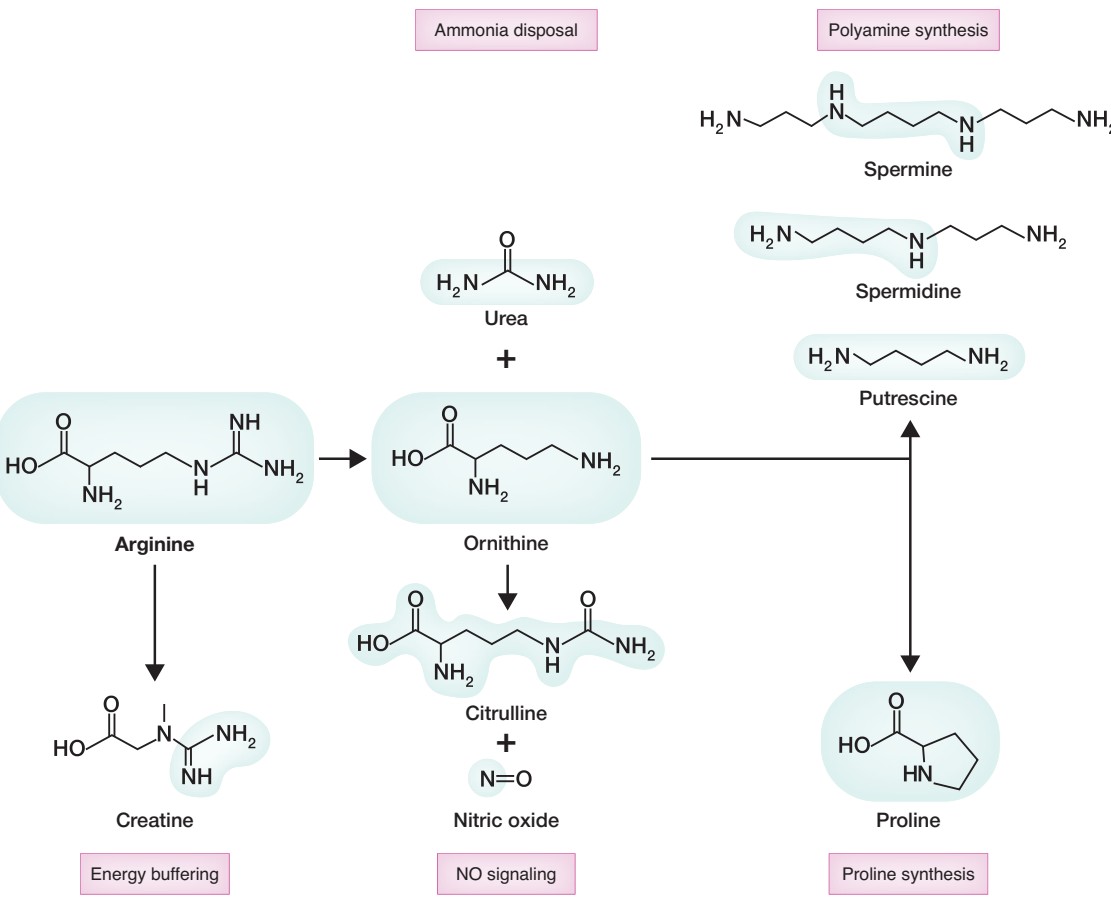

**Figure 3. Metabolites produced from arginine.**

Arginine catabolism contributes to numerous metabolic pathways. Arginine is a substrate for NO synthesis, polyamine production and urea formation. It is also a building block for creatine and proline synthesis. Chemical elements derived from arginine are highlighted in blue. NO nitric oxide.

## Urea and $NH_4^+$ detoxification

$NH_4^+$ is a by-product primarily of mitochondrial glutamine metabolism or the degradation of proteins and amino acids. The transfer of $NH_4^+$ into the urea cycle through mitochondrial CPS1 is the major nitrogen detoxification pathway. Other pathways can also sequester $NH_4^+$. One group recently described how $NH_4^+$ produced by the deamination of glutamine by GLS1 can be recycled and reincorporated into α-KG to produce glutamate through GDH activity (Spinelli et al, 2017). Mitochondrial glutamate synthesis can then serve to modulate $NH_4^+$ levels by producing a nitrogen reservoir to support the synthesis of nonessential amino acids, nucleotides and phospholipids. The outer mitochondrial membrane (OMM) protein glutamine synthetase (GS) also contributes to nitrogen assimilation by utilizing $NH_4^+$ to convert glutamate back to glutamine (Lyu et al, 2024; Zhou et al, 2020).

Interestingly, the tumor suppressor p53 represses the transcription of the urea cycle genes encoding CPS1, OTC, and ARG1. This prevents the elimination of $NH_4^+$ through urea production and also suppresses de novo arginine production (Li et al, 2019). Conversely, in p53-depleted tumor cells, the same urea cycle genes are now upregulated. Thus, p53 plays a potential role in regulating nitrogen reincorporation versus elimination, in addition to its known role in cell cycle control.

Not all animals use arginine-derived urea to secrete excess $NH_4^+$. Marine animals like bony fishes secrete $NH_4^+$ directly into the aquatic environment, while some land-based animals such as mammals secrete excess $NH_4^+$ in the form of urea. Others such as birds and reptiles eliminate excess nitrogen as uric acid. Different species appear to have adopted their nitrogen disposal activity based on water consumption requirements. Approximately 400 mL of water is needed to excrete 1 g of $NH_4^+$, 40 mL of water is utilized in secreting 1 g of nitrogen as urea and only 8 mL of water is required for removal of 1 g of nitrogen as uric acid (Salway, 2018). Therefore, birds and reptiles tolerate dry environments for longer periods without becoming dehydrated. For example, the uric acid cycle helps conserve water during long-distance flights made by migratory birds. Finally, for animals such as marine cartilagenous fishes, the coelacanth and amphibians, urea produced from a functional urea cycle serves as a vital balancing osmolyte and is not considered to be a waste by-product (Withers, 1998).

## Creatine

Arginine is a required precursor in the production of creatine, the major cellular phosphagen that is used to help maintain stable ATP levels in mammalian cells. The synthesis of creatine also requires glycine and three enzymes: L-arginine:glycine amidinotransferase (GATM), methionine adenosyltransferase (MAT) and guanidinoacetate methyltransferase (GAMT). The entire glycine molecule is incorporated into creatine and the amidino group is supplied by arginine. Surprisingly, creatine synthesis consumes some 20–30% of the body's daily intake/production of arginine (Brosnan et al, 2011). Although ATP is the most well-known bioenergetic molecule, it is not a highly diffusible molecule because of its negative charge (Fedosov, 1994; Zala et al, 2017). To address this bioenergetic distribution problem, most mammalian cells possess a creatine shuttle (Fig. 4) that distributes ATP equivalents from mitochondria to sites of ATP utilization in the cytoplasm. Creatine-dependent shuttling of ATP equivalents depends on the activity of mitochondrial creatine kinase (CKMT) and the cytosolic creatine kinases (CKM in muscle cells and CKB in most other cell types). CKMT is localized to the mitochondrial intermembrane space (IMS) (Hung et al, 2014); while CKB and CKM are localized to the cytosol, although a recent study found that ~10% of total CKB can be mitochondrially-associated in brown adipocytes (Rahbani et al, 2021). ATP-dependent creatine phosphorylation provides cells with a way to store ATP-equivalents and to deliver them rapidly from sites of ATP generation (glycolysis and/or OXPHOS) to sites of ATP utilization. Creatine phosphate diffuses through the cytosol over tenfold faster than ATP. As a phosphagen, phosphocreatine is an excellent way to store ATP equivalents as it can be used to recharge ADP to ATP locally in a single-step reaction catalyzed by a creatine kinase (Hargreaves and Spriet, 2020). In cells that rely on OXPHOS for their bioenergetic support, CKMT produces phosphocreatine from the high ratio of ATP to ADP in the mitochondrial IMS and CKM/CKB uses phosphocreatine to convert ADP back to ATP at sites of high ATP utilization in the cytosol. In skeletal muscles and the heart, where ATP demand is high, healthy cells can accumulate a large cytosolic phosphocreatine pool of up to 30 mM (Wallimann et al, 2011).

Cancer cells take advantage of the creatine shuttle for their own proliferation and survival (Kazak and Cohen, 2020). A recent study reported that pancreatic tumor cells upregulate CKB protein expression and shunt available arginine into creatine production to metabolically support cell migration and invasion (Papalazarou et al, 2020). The use of the creatine shuttle to maintain ATP levels during migration might also explain why cytotoxic T cells depend heavily on OXPHOS to infiltrate solid tumors (Simula et al, 2024). Additionally, studies have found that CKB function is required for T cell activation and viability (Di Biase et al, 2019; Samborska et al, 2022), suggesting that energy-demanding processes such as migration require the rapid resupply of ATP at sites of utilization distant from the sites of ATP production.

Interestingly, in bacteria and some invertebrates, arginine itself is the major cellular phosphagen for energy buffering that helps maintain ATP levels. These organisms utilize a phosphoarginine-arginine kinase shuttle instead of the creatine shuttle to maintain their ATP levels (Zhang et al, 2020). Expressing an arginine kinase from *Limulus polyphemus* into both budding yeast (Canonaco et al, 2002) and *E. coli* (Canonaco et al, 2003) improved cell survival

during transient periods (~1 h) of pH stress or nutrient starvation. Therefore, arginine and its derivative creatine have been used widely in evolution to maintain cellular bioenergetics during stress or high ATP demand. However, the mammalian use of creatine-phosphate as a rapidly diffusible phosphagen comes at a cost. Up to 10% of creatine spontaneously cyclizes to creatinine and is secreted in the urine on a daily basis (Wyss and Kaddurah-Daouk, 2000). This is the second highest use of arginine after protein synthesis in humans.

## Nitric oxide (NO)

The sole source of NO in mammalian cells is the catabolism of arginine by one of the family of nitric oxide synthases (NOS). NOS catalyzes the oxidation of arginine to NO and citrulline. As one of the most important messenger molecules in biology, the full description of NO's function can be found in other literature (Lundberg and Weitzberg, 2022). Briefly, NO is a vasodilator in smooth muscle cells, it works as a retrograde neurotransmitter in synapses, and NO is secreted by myeloid cells to kill microbial pathogens (Keshet and Erez, 2018).

## Agmatine

Agmatine is a unique metabolite derived from arginine and its role in cellular homeostasis is not well understood. Agmatine has been best studied in neurobiology because of its role as a neuromodulator through its ability to bind to glutamatergic N-methyl-D-aspartate (NMDA) receptors in the central nervous system (Barua et al, 2019). Inside cells, agmatine is synthesized from arginine by arginine decarboxylase (ADC). There is evidence suggesting that agmatine can be toxic for non-excitable cells and ADC is being tested as a possible cancer therapeutic to deplete arginine in the TME (Patil et al, 2016). Agmatine catabolism also supplies an alternative pathway to produce polyamines from arginine (Lenis et al, 2017).

## Polyamines

Polyamines are polyvalent organic cations that are essential for all living organisms. The dietary uptake of polyamines is substantial and many tissue culture media, as well as serum, contain abundant amount of the major polyamines. All three of the major polyamines—putrescine, spermidine, and spermine—can be synthesized from arginine and these polyamines exist in mammalian cells at mM concentrations (Park and Igarashi, 2013). Polyamines are among the most nitrogen rich and highly charged aliphatic compounds that exist in the human body.

Spermidine is a substrate required for hypusination of a conserved lysine residue in eukaryotic initiation factor 5A (eIF5A) (Park et al, 1981), which drives gene expression that is vital for cell proliferation (Xuan et al, 2023) and differentiation (Puleston et al, 2021). Besides regulating eIF5A, the roles of polyamines on chromatin structure (Park et al, 2023) and histone modifications (Emmons-Bell et al, 2024) are essential and just beginning to be fully appreciated.

At present, it is unclear what proportion of cellular polyamines is produced through arginine catabolism. The major precursor for polyamines is ornithine. The production of polyamines, from

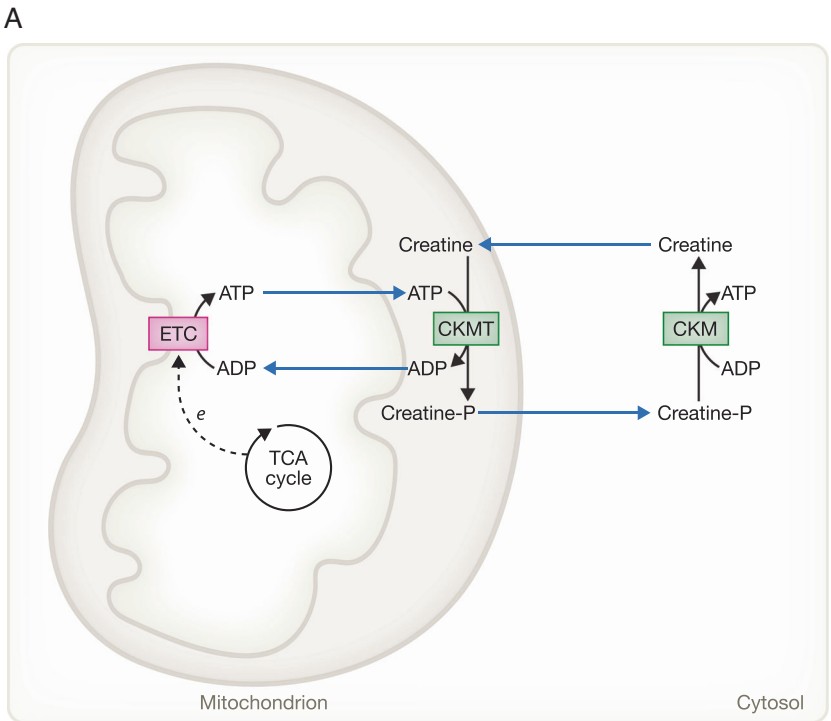

**Figure 4. Phosphagen shuttles for energy buffering and distribution.**

(A) ATP is produced in mitochondria through OXPHOS, which links the TCA cycle and the electron transport chain (ETC) together. Once generated, ATP can be used to phosphorylate creatine through mitochondrial creatine kinase (CKMT) in the mitochondrial intermembrane space to produce phosphocreatine (Creatine-P). Creatine-P, in turn, rapidly diffuses throughout the cytosol, where it donates its phosphate group back to ADP by cytosolic creatine kinase (CKM) to replenish the local ATP levels at sites of high ATP utilization. Creatine then returns to mitochondria where it can be re-phosphorylated to restart the cycle. Therefore, the creatine/creatine-P system serves in both energy transportation and as an energy reservoir to meet ATP demands of a cell. (B) Mammals utilize creatine/creatine-P in the homeostasis of cellular bioenergetics, while bacteria and invertebrates express arginine kinases that convert arginine into phosphoarginine (arginine-P) as their phosphagen shuttle pathway. Black arrows: single biochemical reaction. Blue arrows: translocation of metabolites across membranes. Dotted arrow: reduced substrates from the TCA cycle donate their electrons to the ETC for ATP production. Creatine pathway enzymes are in green and the ETC, comprising of five complexes is in magenta for (A). Chemical formula of creatine/creatine-P and arginine/arginine-P are shown in (B). ATP adenosine triphosphate, ADP adenosine diphosphate, CKMT mitochondrial creatine kinase, CKM creatine kinase M-type, *e* electron transfer, Arginine-P phosphoarginine, Creatine-P phosphocreatine, OXPHOS oxidative phosphorylation, TCA cycle tricarboxylic acid cycle, ETC electron transport chain.

arginine is nitrogen-depleting. Arginine conversion to the poly-amine precursor ornithine generates an equimolar amount of urea which must be secreted as waste. The resulting ornithine is converted into putrescine by the rate-limiting enzyme ornithine decarboxylase (ODC). Interestingly, ODC is an important gene target of the MYC transcription factor (Bachmann and Geerts, 2018). MYC amplification is an oncogenic driver leading to ODC overexpression and polyamine accumulation in tumor cells (Asai et al, 2018; Geck et al, 2020; Kingsnorth et al, 1984). At the same time, MYC activation drives both mitochondrial glutaminolysis (Edwards-Hicks et al, 2022; Wise et al, 2008) and glutamate-dependent mitochondrial synthesis of ornithine and polyamines (Chalishazar et al, 2019; Tsai et al, 2012).

While both glutamine and arginine can be used to produce ornithine, glutamine-dependent ornithine production conserves reduced nitrogen. In contrast, arginine-dependent ornithine production leads to the loss of two molecules of reduced nitrogen as urea. Still, most reports concerning ornithine/polyamine production focus on arginine as the primary precursor, while glutamine is considered a secondary source of ornithine. This is puzzling as glutamine is present in extracellular fluid at 500 μM and arginine is present at 50 μM. Indeed, pancreatic cancer cells were reported to utilize glutamine for ornithine production to support their polyamine biosynthesis only as arginine becomes limiting in the TME (Lee et al, 2023).

Extracellular polyamines from the diet enter cells through polyamine permeases or through receptor-mediated endocytosis (Holbert et al, 2022). The coordination between arginine/gluta-mine-dependent polyamine synthesis and exogenous polyamine uptake is not known.

In addition to their nuclear/cytosolic functions, polyamines also contribute to mitochondrial fitness. Polyamine-mediated activation of eIF5A-dependent translation stimulates OXPHOS across multiple cell types (Liang et al, 2021; Puleston et al, 2019; Zhou et al, 2022). In addition, spermidine can directly bind to mitochondrial proteins, promoting fatty acid oxidation and mitochondrial respiration (Al-Habsi et al, 2022).

### Proline

Proline is another potential product of arginine catabolism (Fig. 1). Proline is the rate-limiting amino acid required for the synthesis of collagen and other extracellular matrix proteins (Guillard and Schwörer, 2024). The addition of proline into proteins is slower than other amino acids (Huter et al, 2017; Melnikov et al, 2016), and the addition of consecutive prolines to produce collagen can lead to ribosomal stalling (Gutierrez et al, 2013; Ude et al, 2013). This delay can be circumvented through hypusinated eIF5A elongation factor, which promotes productive positioning of proline-tRNA during peptide bond formation (Barba-Aliaga et al, 2021).

Besides protein synthesis, proline production helps regulate cellular redox balance (Guillard and Schwörer, 2024; Zhen et al, 2024). Mitochondrial pyrroline 5-carboxylate reductase 1 (PYCR1) consumes NADH to convert P5C into proline (Westbrook et al, 2022). P5C can be supplied either through arginine catabolism into ornithine and subsequently to GSA/P5C through OAT activity; or through glutamate reduction into GSA/P5C by P5CS using mitochondrial NADPH (Linder et al, 2023; Niu et al, 2023; Zhong et al, 2022).

At present, how much of the proline synthesis comes from de novo synthesis from glutamate or from the catabolism of arginine during cell proliferation, inflammation and wound healing remains to be determined (Schwörer et al, 2020). In tissue culture cells, arginine is reported to be a contributing source of proline production (Geiger et al, 2016) but this may be due to the non-physiologic levels of arginine present in most tissue culture media. Instead, recent studies suggest that glutamine is the major source of proline biosynthesis (Ryu et al, 2024; Schwörer et al, 2020; Tran et al, 2021; Zhu et al, 2021).

Arginine-derived proline has been proposed to be a bioenergetic substrate for mitochondrial ATP production because there are mitochondrial enzymes that convert proline to glutamate through proline catabolism that is sequentially catalyzed by proline dehydrogenase (PRODH) and P5C dehydrogenase (P5CDH) (Phang, 2019). Interestingly, the upregulation of these enzymes in response to nutrient starvation has been reported to be p53-dependent (Lacroix et al, 2020; Scott et al, 2019; Yoon et al, 2004). With the recent discovery of macropinocytosis as an alternative mechanism for the uptake of nutrients to support starving cells (Puccini et al, 2022), it appears that protein-incorporated proline can be an additional source to support mitochondrial OXPHOS and reductive biosynthesis. The most common amino acids in extracellular proteins such as collagen are glycine and proline (Kay et al, 2021).

## Arginine in protein synthesis and post-translational modifications of residues

Inside the cell, the arginine side chain is always positively charged (high pKa of ~13.8) (Fitch et al, 2015). This means that arginine as a metabolite cannot permeate membranes freely (Topal et al, 2006) and movement between cellular compartments requires cationic amino acid transporters (CATs) (Pizzagalli et al, 2021).

The arginine side chain remains protonated even if it is buried within the hydrophobic cores of proteins (Harms et al, 2011) or embedded into lipid membranes (Li et al, 2013). This constitutive positive charge is vital for arginine contributing to protein function. For example, arginine serves as a voltage-sensing residue in sodium and potassium ion channels (Armstrong et al, 2016; Tombola et al, 2005). Surprisingly, mitochondrial (Fukasawa et al, 2015; von Heijne, 1986) and nuclear targeting sequences (Requião et al, 2017) are both enriched in arginine. Mutations to arginine residues in the mitochondrial targeting sequence of either malate dehydrogenase (Chu et al, 1987) or pyruvate dehydrogenase E1 alpha (Takakubo et al, 1995) significantly impair mitochondrial import.

In addition, arginine-rich motifs facilitate protein binding to negatively charged nucleic acids for DNA binding, RNA processing or translocation of RNA/DNA into and around the cytosol (Bayer et al, 2005; Hase et al, 2020). In some RNA-binding proteins, RNA recognition domains contain intrinsically-disordered regions (IDRs) that are arginine-rich (Wang et al, 2018). The presence and amount of arginine in IDRs regulate the ability of RNA-binding proteins to phase separate and form liquid-like condensates. Hence, arginine (along with tyrosine) repeats in an amino

**Figure 5. Post-translational modifications on arginine residues.**

The methylation of arginine residues is catalyzed by the protein arginine methyltransferases (PRMTs). PRMTs of Type I, II and III can all produce monomethylarginine (MMA). In addition, Type I add other methyl groups to the same nitrogen atom to yield asymmetric dimethylarginine (ADMA), while Type II PRMTs attach the second methyl group to the other N-terminal nitrogen of the arginine residue, forming symmetric dimethylarginine (SDMA). Type III PRMT is only capable of mono-methylation. In contrast, peptidylarginine deiminases (PADs) performs the hydrolysis of the guanidino group to form citrulline.

acid sequence provide a molecular signature to predict the phase separation properties of a protein (Wang et al, 2018).

Given the importance of arginine residues in proteins, arginine deprivation can have dramatic effects on protein translation. Indeed, limiting arginine causes a rapid reduction in charged arginine transfer RNAs (tRNA) and induces ribosomal stalling over arginine codons (Darnell et al, 2018; Hsu et al, 2023). In lung cancer cells, arginine deprivation can result in the substitution of cysteine for arginine because of tRNA misalignment during translation. The resultant increase in arginine-to-cysteine substitutions in the cellular proteome elevates the cell's capacity to buffer reactive oxygen species and promotes resistance to oxidative stress (Yang et al, 2024). In addition, uncharged arginine tRNAs activate the GCN2 amino acid sensor, which mediates downstream ATF4-driven stress response, promoting cell arrest and quiescence (Missiaen et al, 2022).

## Arginine undergoes post-translational modifications (PTMs) and epigenetic regulation

Protein arginine methyltransferases (PRMTs) catalyze the methylation of the guanidino group on arginine residues incorporated into histones and other proteins (Hwang et al, 2021) (Fig. 5). Methylation of the side chain does not alter the net positive charge of arginine but instead alters the charge distribution. Overall, this modification increases the net hydrophobicity of the protein (Wu

et al, 2021b). The PRMT family of enzymes can be divided into three categories (Yang and Bedford, 2013): (1) Type I enzymes di-methylate their substrates asymmetrically. As a result, both methyl groups are presented on the same guanidino nitrogen atom. (2) On the other hand, type II enzymes are symmetrical dimethyl-transferases. Type II enzymes transfer one methyl group each to the two guanidino nitrogen atoms. (3) In contrast, the sole member of type III PRMT can only monomethylate arginine residues.

Histones are an important target of PRMTs. The best-studied methylations occur on the N-terminal regions of histones H3 and H4, including H3R2, H3R17, and H4R3 (Fulton et al, 2018). Histone methylation results in chromatin restructuring and RNA transcriptional changes. Besides histone substrates, PRMTs methylate many other targets. This is especially important for proteins that can bind RNA, since arginine methylation weakens the interactions of arginine-containing proteins with nucleic acids (Wu et al, 2021b). At present, there are thousands of reported arginine methylated proteins characterized on the PhosphoSite Plus database (Hornbeck et al, 2015; Larsen et al, 2016).

As such, a relationship between PRMTs and cancer is starting to emerge. Indeed, the dysregulation of PRMTs can drive oncogenic pathways in multiple cancer types (Hing et al, 2023; Liu et al, 2016; Zhao et al, 2019). Targeting PRMTs with small molecule inhibitors is being explored as a clinical strategy in cancer treatment (Guccione et al, 2021). Besides cancer, PRMT activity correlates

with other pathologies, such as cardiovascular diseases and neurodegenerative disorders (Couto et al, 2020).

Protein citrullination is another PTM on protein-incorporated arginine residues. This modification is catalyzed by the calcium-regulated peptidylarginine deiminase (PAD) family of enzymes and results in the conversion of peptidyl arginine to peptidyl citrulline. In this case, the net positive charge of the side chain is lost (Fuhrmann et al, 2015) (Fig. 5). The decrease in net charges of PAD-targeted proteins alters protein folding and can even cause denaturation due to changes in intra- and inter-molecular ionic interactions (Cau et al, 2018; Witalison et al, 2015). Like arginine methylation, protein citrullination plays an important role in both normal development (Christensen et al, 2022) and in carcinogenesis (Yuzhalin, 2019). The conversion of arginine to citrulline by deiminase is also being explored therapeutically. Bacteria secrete arginine deiminases to modulate the arginine concentration in the environment (Novák et al, 2016). In vivo, administration of *Mycoplasma*-derived arginine deiminase is now being studied as a potential cancer therapy (Riess et al, 2018).

# Cellular acquisition of arginine

Multiple pathways are deployed by cells to acquire arginine. These include: (1) De novo synthesis as described above, (2) direct uptake from the extracellular environment through amino acid transporters, and (3) degradation of extracellular proteins taken up by macropinocytosis (Fig. 6).

## Direct uptake through amino acid transporters

Most mammalian cells do not synthesize sufficient de novo arginine to meet their demand (Morris, 2016). Uptake of extracellular arginine is mediated by members of the solute carrier (SLC) 7 family of cationic amino acid transporters. The SLC7 family consists of 13 members, which are divided into two subfamilies: the cationic amino acid transporters (CATs) with four members (SLC7A1-4); and the L-type amino acid transporters (LATs) with the other nine members (SLC7A5-13). Structurally, CATs and LATs are different (Pizzagalli et al, 2021). Besides taking in arginine, SLC7 transporters can also bring in products of arginine metabolism. The SLC7A5 transporter has recently been reported to bring in exogenous citrulline when arginine becomes limited (Dunlap et al, 2025). The importance of extracellular arginine to support cellular demand cannot be overstated. Multiple cancer cells exhibit upregulation of arginine transporters to support their metabolic demands (Apiz Saab et al, 2023; Missiaen et al, 2022).

## Acquiring arginine through macropinocytosis

Macropinocytosis provides an alternative mechanism through which cancer cells (Palm, 2019) and immune cells (Charpentier et al, 2020) acquire nutrients from the extracellular environment. This mechanism of nutrient uptake is particularly important in solid tumors, where nutrients are scarce. Indeed, arginine is one of the most limited amino acids in the TME of pancreatic tumors (Apiz Saab et al, 2023; Lee et al, 2023). Macropinocytosis of extracellular proteins and the subsequent degradation of those

proteins in the lysosome is a major route of amino acid uptake from the TME (Commisso et al, 2013). Lysosomal arginine accumulation is a critical regulator of the release of amino acids from the lysosome into the cytosol. During cell growth supported by macropinocytosis, lysosomal arginine acts as an allosteric regulator of the SLC38A9 which promotes an efflux of lysosomal amino acids into the cytoplasm (Lei et al, 2018).

What regulates how cells switch between SLC7-mediated arginine uptake and macropinocytotic acquisition of arginine is at present unknown, but it might be cell type dependent. A recent study suggested that pancreatic ductal adenocarcinoma (PDAC) cells upregulate cell surface amino acid transporters, rather than macropinocytosis, to acquire exogenous arginine when de novo arginine synthesis pathways are impaired (Apiz Saab et al, 2023). In contrast, PDAC cells stimulate macropinocytosis through EGFR-Pak signaling when they are glutamine deprived and endogenous asparagine synthesis is compromised (Lee et al, 2019). Sarcoma tumor cells, on the other hand, have been reported to activate macropinocytosis when they are arginine-deprived (Rogers et al, 2023). A possible route for how arginine may stimulate macropinocytosis is through activation of GCN2 (Hamanaka et al, 2024; Missiaen et al, 2022), whose activation suppresses mTOR activity to enhance macropinocytosis and upregulate lysosomal proteins such as cathepsin L to degrade captured proteins (Nofal et al, 2022).

# Endogenous arginine synthesis

Many tumors exhibit dysregulation in the expression of urea cycle enzymes. This decreased expression can result in a rewiring of nitrogen metabolism in cancer cells (Ochocki et al, 2018), enhancing the ability to meet the increased demand for nucleotides and amino acids (Lee et al, 2018). Disruption of the proximal enzymes in the urea cycle can impair a cell's ability to synthesize arginine. Clinically, tumor arginine auxotrophy is correlated with poor prognosis and chemoresistance (Riess et al, 2018). Tumors where more than 50% of the cases are auxotrophic for arginine include: head and neck squamous cell carcinoma, clear cell renal cell carcinoma (ccRCC), malignant melanoma, prostate, breast, pancreatic, acute myeloid leukemia, malignant pleural mesothelioma and most surprisingly hepatocellular carcinoma (HCC)—with 100% of cases found to be arginine auxotrophs. In contrast, small cell lung cancer and glioblastoma multiforme are auxotrophic for arginine 44% and 20% of the time, respectively (Riess et al, 2018).

The most common mechanism that mediates arginine auxotrophy appears to be silencing of ASS1. A publicly available database, The Cancer Genome Atlas has been used to assess gene expression patterns of ASS1 for given types of cancer and these levels were found to correlate with the extent of arginine auxotrophy (Cheng et al, 2018; Ding et al, 2023; Keshet et al, 2020; Lee et al, 2018). In most cases, the lack of ASS1 expression is often associated with hypermethylation of the ASS1 promoter (Szlosarek et al, 2006) but HIF1α-induced microRNAs have also been found to transcriptionally repress ASS1 mRNA (Silberman et al, 2019). A potential explanation for why cancer cells repress ASS1 and become dependent on an exogenous source of arginine is that ASS1 consumes aspartate. Aspartate is a critical amino acid for nucleotide synthesis in support of cell growth (Sullivan et al, 2018). Indeed, silencing ASS1 in cancer cells has been shown to suppress

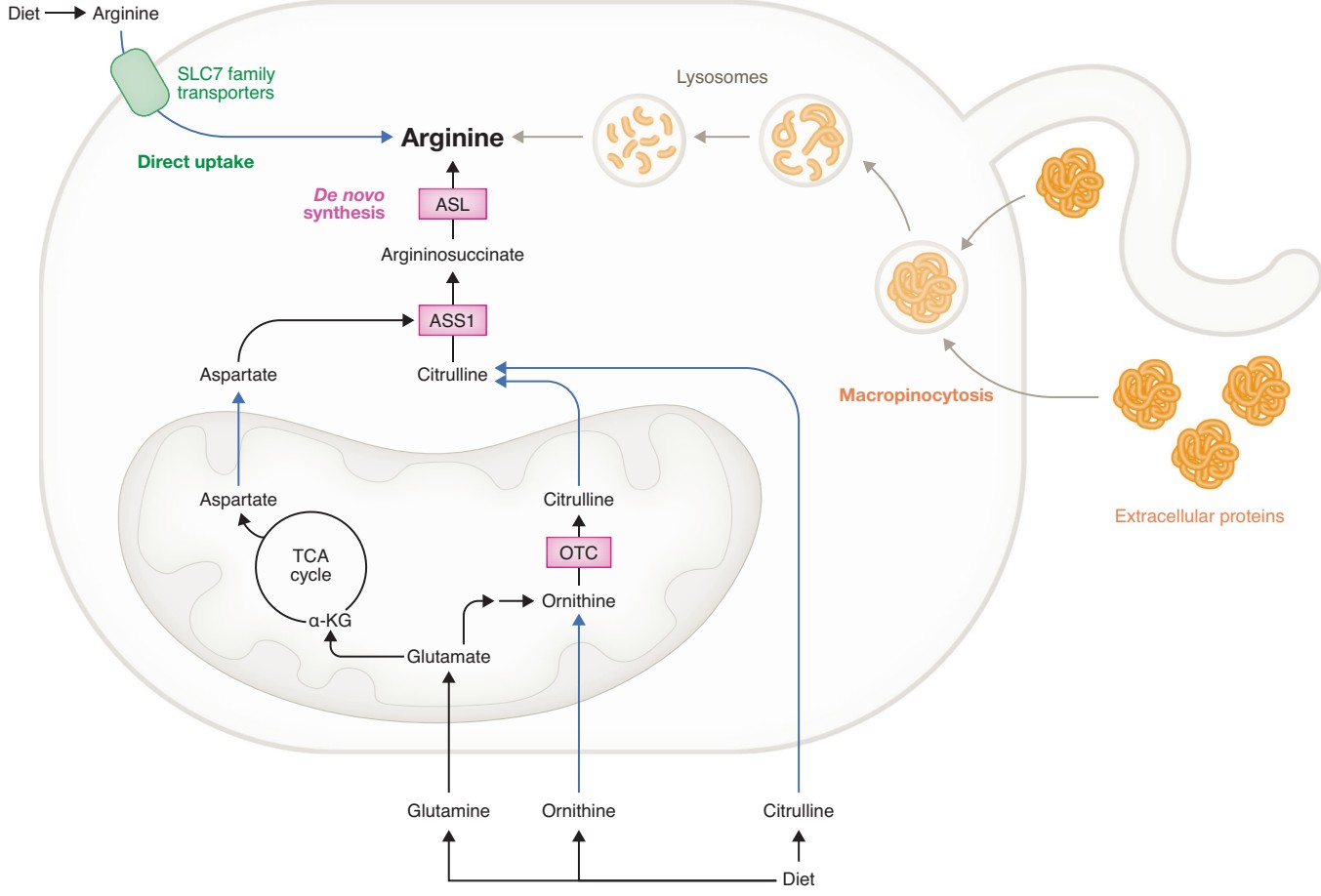

**Figure 6. Arginine acquisition pathways.**

As a semi-essential amino acid, intracellular arginine level can be maintained in three different ways (Apiz Saab et al, 2023). Arginine can be directly taken up from the extracellular environment through the SLC7 family of transporters. In addition, extracellular proteins and debris can be scavenged through macropinocytosis. Once internalized, macropinsomes are sorted and trafficked, and ultimately merged with lysosomes. Captured cargo are then degraded within the lysosomal lumen and the resultant amino acids (such as arginine) can be released into the cytosol or re-distributed into other cellular compartments such as mitochondria. Lastly, dietary supply of glutamine, ornithine and citrulline can all be used as substrates for de novo synthesis of arginine through the urea/citrulline cycles. Black arrows: single biochemical reaction. Blue arrows: translocation of metabolites across cellular compartments. Grey arrows: Capture and lysosomal degradation of extracellular proteins. OTC ornithine transcarbamylase, ASS1 argininosuccinate synthase 1, ASL argininosuccinate lyase, α-KG α-ketoglutarate, TCA cycle tricarboxylic acid cycle.

the urea cycle and redirect available aspartate into pyrimidine biosynthesis (Rabinovich et al, 2015). ASS1-depleted breast cancer cells are unable to produce aspartate endogenously and have impaired growth when subjected to arginine deprivation (Cheng et al, 2018). There is also evidence suggesting that ASS1 is a tumor suppressor through the ability of ASS1 to promote p53 activity (Lim et al, 2024), or activate an ER stress response independent of its role as a metabolic enzyme (Kim et al, 2021).

Thus, the role of the urea cycle and urea cycle enzymes in the pathogenesis and progression of cancer may be context-dependent. A proliferating cell's need to suppress the urea cycle and their need for arginine can be in conflict. For example, in both ccRCC (Ochocki et al, 2018) and HCC (Missiaen et al, 2022; Mossmann et al, 2023) that express low levels of ASS1, the cancer cells concomitantly repress arginases (ARG1/2) and polyamine synthesis enzyme agmatinase (AGMAT) to maintain a high intracellular arginine concentration after assimilating arginine from the extracellular environment. Other types of cancer cells that

experience arginine starvation adapt by re-expressing ASS1 (Crump et al, 2021; Tsai et al, 2012). Interestingly, there are even cancer cell lines that rather than repress ASS1, upregulate ASS1 expression. High ASS1 promotes arginine synthesis and NO signaling (Keshet et al, 2020) and confers resistance to ferroptosis by modulating the TCA cycle and glutamine metabolism (Hu et al, 2023; Keshet et al, 2020).

## Extracellular arginine levels are low in tumors and inflammatory lesions

Immune cells have a strong dependence on arginine and extracellular arginine is frequently depleted in inflammatory lesions, wounds, and tumors. This depletion occurs because of increased arginine consumption by cells engaged in growth, repair, or as a result of ARG1 secretion into the environment from macrophages (Menjivar et al, 2023; Sosnowska et al, 2021) (Fig. 7). The absolute level of

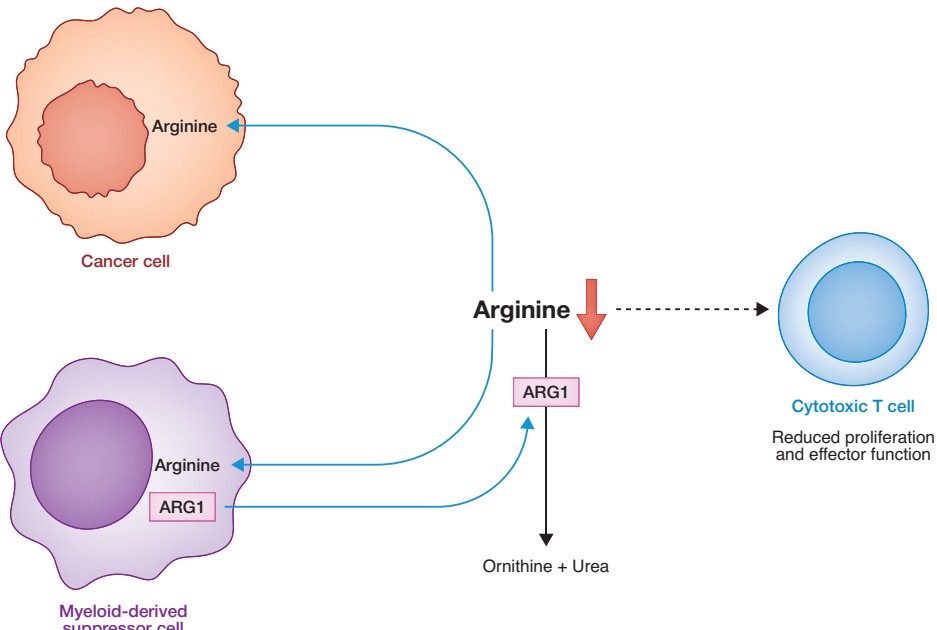

**Figure 7. Cell–cell competition for arginine in the tumor microenvironment.**

In the tumor microenvironment (TME), infiltrating cytotoxic T cells face competition for key nutrients such as arginine. Cancer cells which are often auxotrophic for arginine taking in large amounts of arginine to maintain their proliferation and cell viability, which creates an immunosuppressive environment due to reduced arginine (Mussai et al, 2015). To make matters worse, immunosuppressive macrophages also take up arginine (Raber et al, 2012; Tharp et al, 2024) and secrete ARG1 (Rodriguez et al, 2004; Sosnowska et al, 2021) into the extracellular environment, hydrolyzing free arginine which further reduces nutrient availability in the TME. Cytotoxic T cells are highly susceptible to arginine deprivation, leading to OXPHOS inhibition (Crump et al, 2021; Fletcher et al, 2015), hampered glycolysis (Crump et al, 2021) and a reduction in cytokine production (Choi et al, 2009), proliferation (Czystowska-Kuzmicz et al, 2019), differentiation (Werner et al, 2016) and CD3ξ expression (Rodriguez et al, 2003). Black arrow: single biochemical reaction. Blue arrows: translocation of metabolite/enzyme across cellular compartments. ARG1 arginase 1.

extracellular arginine in tumors is controversial. It remains a technical challenge to accurately measure either the extracellular or intracellular concentration of arginine (Verma et al, 2017). Some arginine biosensors have been reported (Hartenbach et al, 2007; Rogers et al, 2021) but overall, real-time, dynamic measurements of arginine availability continue to be a work in progress. What exists in the literature does not support the conclusions being drawn from cells cultivated in standard tissue culture media containing 0.4–1.1 mM arginine. For example, in the TME of murine PDAC tumors, arginine was found to be present at 2–5 µM compared to 50–120 µM in serum (Apiz Saab et al, 2023; Lee et al, 2023; Sullivan et al, 2019). In vascularized tumors such as renal cell cancer, samples taken from patients has revealed that the tumor interstitial fluid (TIF), adjacent normal kidney interest fluid (KIF) and the plasma serum all have similar concentrations of arginine that ranges from 50–100 µM (Abbott et al, 2024). Citrulline can also be taken up by cancer cells and used to support de novo arginine synthesis, and citrulline concentration in the TIF was found to be subsequently lower (10 µM) than in serum (38 µM).

Thus, the physiologic levels of arginine present in the TME are far from the concentrations of arginine found in tissue cell culture medium. The two most commonly used cell culture media are Dulbecco's modified Eagle's medium (DMEM) and Roswell Park Memorial Institute (RPMI) medium. DMEM contains 400 µM of arginine while RPMI contains 1.14 mM of arginine. Arginine is supplemented in tissue culture media to a much greater extent than

all other amino acids with the exception of glutamine. Therefore, there is a danger that under these supraphysiological concentrations of arginine, cells come to depend on arginine in non-physiologic ways, masking key aspects of both physiologic and pathophysiologic arginine metabolism.

For example, intracellular arginine levels have been shown to regulate mTOR signaling. The mammalian target of rapamycin TOR (mTOR) is a protein kinase that stimulates multiple anabolic processes including protein translation and lipid synthesis (Laplante and Sabatini, 2009). Signaling through the mTOR pathway is conserved from yeast to metazoans (Zaman et al, 2008). mTOR activation is under regulation by various upstream signals, including amino acid availability (Shimobayashi and Hall, 2016). A detailed description of the different factors involved in mTOR activation is beyond the scope of this review, but other literature can be consulted (Goul et al, 2023).

mTOR complex 1 (mTORC1) activation takes place on the cytoplasmic face of the lysosomal membrane, driven by RAG GTPases and the Ragulator (Rag-Ragulator) complex that are in close proximity to the vacuolar $H^+$-ATPase (Betz and Hall, 2013). When nutrients are scarce, mTORC1 recruitment to the lysosome is inhibited because of the suppression of its upstream positive regulator GATOR2. When bound to Sestrin2 and/or CASTOR1, GATOR2 activity is inhibited (Valenstein et al, 2022). When intracellular arginine increases, however, the metabolite binds directly to CASTOR1 and suppresses CASTOR1-GATOR2 interaction (Chantranupong et al, 2016). Similarly, intracellular leucine binds to Sestrin2, and subsequent

dissociation of Sestrin2 from GATOR2 also contributes to mTORC1 activation (Wolfson et al, 2016).

The accumulation of lysosomal arginine also regulates amino acid release from lysosomes and helps maintain cytosolic amino acid levels and mTOR activity. SLC38A9 is a sodium-coupled amino acid transporter localized to the lysosomal membrane (Rebsamen et al, 2015). When lysosomal arginine binds to SLC38A9, a conformational change is induced, opening the channel for efflux of amino acids from the lysosomal lumen into the cytosol (Lei et al, 2021). Among other essential amino acids, SLC38A9 transports leucine. Once released into the cytosol, leucine can sustain Sestrin2-dependent activation of mTORC1. Arginine binding also promotes SLC38A9 interaction with the Rag-Ragulator complex to promote mTORC1 activation (Wyant et al, 2017). Thus, the presence of arginine activates mTORC1 through two different pathways, both by binding CASTOR1 in the cytosol and by binding SLC38A9 on the lumenal side of lysosomes.

Arginine-mediated mTORC1 activation is also reported to stimulate translocation of the TEAD4 transcription factor into the nucleus to upregulate nuclear-encoded OXPHOS genes (Chen et al, 2021; Kumar et al, 2018) and promote mitochondrial fitness. This may help explain the reported ability of arginine to stimulate OXPHOS and promote immune cell fitness (Geiger et al, 2016). Since arginine signaling involves both mitochondria and lysosomes, lysosomal release of stored amino acids may also promote amino acid import into mitochondria to support mitochondrial function. iPSC-derived dopaminergic neurons harboring mutations to Parkin have disrupted mitochondria-lysosome contact sites (Peng et al, 2023). Interestingly, these cells have more arginine accumulated in their lysosomes relative to their mitochondria. Restoring contact sites in these mutant neurons partially rescues the balance of arginine between the two organelles, suggesting a role for mitochondria-lysosome contact sites to facilitate arginine and/or other amino acid transfer.

### Arginine signaling for transcriptional control

Apart from arginine's effects on mitochondria and lysosomes, a recent study showed that arginine binds to the splicing factor RNA-binding protein 39 (RBM39) in hepatocellular carcinoma (HCC) cells (Mossmann et al, 2023). Transformed HCC cells have a silenced urea cycle but they upregulate the cationic amino acid transporter SLC7A1 to scavenge for extracellular arginine (Missiaen et al, 2022). Binding of cytosolic arginine to the N-terminal of RBM39 promotes translocation of the protein into the nucleus and enhances the transcription of multiple metabolic genes. One gene that is upregulated encodes for the asparagine synthetase (ASNS). Increased asparagine production through ASNS activity promotes tumor progression, since asparagine can either be used in protein synthesis for growth (Pavlova et al, 2018) or be exported out of the cell in exchange for extracellular arginine (Krall et al, 2016). Given the critical role of RBM39 in tumorigenesis, a small molecule (indisulam) that targets and degrades RBM39 has been found to be an effective treatment for HCC and other tumor types (Nijhuis et al, 2022; Xu et al, 2021).

## The role of arginine in metastasis

Arginine metabolism has been reported to differ between tumor cells at their primary site versus metastatic sites. This difference remains poorly understood. In esophageal squamous cell carcinoma (ESCC) tumor metastasis (Sun et al, 2024), the metastatic tumors have been reported to have lower arginine and ornithine concentrations compared to the primary tumor. An in-depth survey of the expression levels in urea cycle enzymes also revealed that ASS1 and ASL were both upregulated in the primary tumor relative to the metastatic site. This suggests that de novo synthesis of arginine promotes the accumulated levels of intracellular arginine in the primary site. Strikingly, silencing ASS1 or ASL to suppress de novo arginine synthesis in the primary tumor promoted metastasis. In contrast, overexpressing these enzymes in the primary tumor has the opposite effect, resulting in improved tumor growth but suppressed metastasis. On the other hand, comparison of primary (kidney) and clonally related lung metastases revealed upregulated ASS1 expression in the metastatic cell lines relative to their primary tumor-derived counterparts (Sciacovelli et al, 2022). ASL levels in both sample groups were relatively the same. The ASS1 upregulation in the metastatic kidney cancers conferred resistance to arginine starvation.

## Targeting arginine metabolism in cancer

The semi-essential dependence of tumor cells and immune cells for arginine has led to efforts to decrease the levels of arginine in patients to either directly suppress tumor growth or by enhancing the efficacy of existing therapies. Some therapeutically relevant proteins that are employed to deplete the arginine environment include: ARG1, arginine decarboxylase (ADC) and arginine deiminases (ADI). *Mycoplasma*-derived ADI-PEG20 is most commonly used in research and was reported to be therapy that has the potential to suppress tumor growth in preclinical studies (Riess et al, 2018). Clinical trials aimed at depleting extracellular arginine in ASS1-silenced tumors are underway, with mixed patient outcomes reported. A phase III clinical trial using ADI-PEG20 in HCC failed (Abou-Alfa et al, 2018), while the combination of ADI-PEG20 together with other chemotherapies showed promise for patients with pleural mesothelioma (Szlosarek et al, 2024).

A major stumbling block to arginine-depleting strategies is the ability of cancer cells to adapt to the drop in available arginine. The most direct adaption reported is the re-expression of ASS1 for de novo synthesis of arginine (Crump et al, 2021; Tsai et al, 2009; Tsai et al, 2012). The arginine uptake transporter SLC7A1 (Missiaen et al, 2022) and macropinocytosis (Rogers et al, 2023) are also reported to be upregulated by cancer cells when extracellular arginine is depleted. Some investigators are also exploring the opposite approach. Instead of depleting arginine, attempts are being undertaken to increase the concentration of arginine of the TME to improve cancer immunotherapy. Activated T cells have a high dependency on arginine metabolism (Geiger et al, 2016; Martí i Líndez et al, 2019) and enhancing T cells access to arginine can increase tumor-killing ability. Preclinical studies have shown that infiltrating cytotoxic T cells have improved effector function and cell viability when there is more free arginine available in the immediate environment (Canale et al, 2021; Zhang et al, 2022). In addition, overexpressing urea cycle enzymes (Fultang et al, 2020) and amino acid transporters (Panetti et al, 2023) in chimeric antigen receptor (CAR) T cells might be another way to improve tumor-killing in the TME.

## Arginine as a treatment for other diseases

Although we have focused on the role of arginine in cancer, arginine plays a role in other disease processes. For example, arginine is the only metabolite that can produce NO and vasodilating the blood vessels to improve blood circulation is correlated to improved health. The amount of arginine produced by de novo synthesis alone is not sufficient to support developmental cell growth in vivo, and supplemental arginine is reported to improve development, lactation, and fertility (Wu et al, 2021a). Arginine supplementation has also been found beneficial in patients with mitochondrial diseases where de novo synthesis of arginine is impaired (Argudo et al, 2022). Arginine infusion is an FDA-approved treatment of pre-eclampsia hypertension. Pre-eclampsia is a serious complication in pregnancy, having a prevalence occurring in between 2 and 8% of pregnancies (Dorniak-Wall et al, 2014). Although the cause(s) of pre-eclampsia are not fully elucidated, arginine insufficiency during pregnancy may contribute (Morris et al, 1996). Asymmetric dimethyl arginine, a competitive inhibitor of NOS, has been found to be elevated in women with pre-eclampsia (Savvidou et al, 2003). Arginine is used in pre-eclampsia to boost the systemic arginine levels to counteract this competitive suppression of NOS. Encouragingly, randomized clinical trials have found that arginine can ameliorate the symptoms of pre-eclampsia (Camarena Pulido et al, 2016; Savvidou et al, 2003), even amongst high-risk populations (Vadillo-Ortega et al, 2011).

## Conclusion

Overall, great strides have been made in recent years to better understand the semi-essential role of arginine in health and disease. However, there are still important challenges and unresolved questions concerning arginine metabolism. For example, what are the precise differences or similarities in the roles of cytosolic ARG1 and mitochondrial ARG2? Why do cells produce ornithine and urea in two different compartments? How does intracellular ammonia get partitioned between de novo synthesis of arginine and other amino acids or its elimination through urea production? In terms of nutrient sensing, how does a cell determine whether to use the urea cycle to secrete excess nitrogen, or divert excess nitrogen into creatine synthesis to support bioenergetics, or promote NO production to maintain vascular tone and innate immune function? It remains unclear whether physiologic levels of arginine ever contribute to the production of polyamines and proline in vivo or whether de novo production of polyamines and proline depends exclusively on mitochondrial reductive synthesis from glutamate. All in all, the study of arginine metabolism will remain a vital field that continues to provide new insights into cell biology in the coming decade.

## Peer review information

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

## Acknowledgements

We thank members of the Thompson laboratory, especially Tullia Lindsten and Charles Ng for insightful feedback and comments on the manuscript. Keun Woo Ryu is supported by the Hunter Douglas Fellowship in Breast Cancer Research (13459) and the BRIA Postdoctoral Researcher Innovation Grant (18057). Craig B Thompson is supported by grants from the NCI (P30 CA008748 and R35 CA283988).

## Author contributions

**Tak Shun Fung**: Conceptualization; Resources; Visualization; Writing—original draft; Project administration; Writing—review and editing. **Keun Woo Ryu**:

Conceptualization; Validation; Project administration; Writing—review and editing. **Craig B Thompson**: Conceptualization; Resources; Supervision; Funding acquisition; Validation; Writing—original draft; Project administration; Writing—review and editing.

## Disclosure and competing interests statement

Craig B Thompson is a founder of Agios Pharmaceuticals. He also serves on the Board of Directors of Regeneron and Charles River Laboratories. The other authors declare no competing interests.

