## [Peer Review File · The EMBO Journal]

Arginine: at the crossroads of nitrogen metabolism

Tak Shun Fung, Keun Woo Ryu and Craig Thompson,

Corresponding author: Craig Thompson (thompsonc@mskcc.org)

Review Timeline:

Submission Date:	24th Sep 24
Editorial Decision:	10th Nov 24
Revision Received:	6th Dec 24
Accepted:	10th Dec 24

Editor: Daniel Klimmeck

Transaction Report:

Dear Craig and all,

Thank you again for sending us your review article for consideration, as well as for your patience with our feedback at this time. We have asked two dedicated metabolism - cancer experts to assess your manuscript, and in the meantime got feedback from both of them, which I enclose below.

As you will see, the experts much appreciate the review and find it timely and worth publishing. They also provide constructive feedback on how to further improve it by advancing the discussion, as well as clarifying a number of aspects, and revise the figure work.

I hope you will find the comments helpful. I am sure that an amended version incorporating the suggestions made by the referees will be highly noted and appreciated. I would thus like to invite you to submit such a revised version using the link enclosed below.

Please let me know in case I can be of any help with this.

with
Best wishes,

Daniel

Daniel Klimmeck, PhD
Senior Editor
The EMBO Journal

Referee #1:

This is a well written and very timely review. The content is excellent and synthesizes the findings of a significant amount of literature into a balanced and comprehensive overview of our current understanding of arginine metabolism, especially in the setting of cancer. The paper also proposes important directions for the future to address current gaps in knowledge. I only have a few minor edits:

- 1) Urea Cycle loss was first documented in ccRCC in Ochocki et al. (2018) Cell Metabolism and should be cited as such.
- 2) The figures could be improved with the use of BioRender or a similar platform.
- 3) The sentence starting with "For example, in both HCC.." on page 21 has the two references reversed for the HCC and ccRCC findings.
- 4) On page 22, the sentence "For example, TME in the murine..." appears to be grammatically incorrect and should be edited.

Referee #2:

Fung, Thompson and colleagues present an authoritative review on the biology of arginine in mammals, with a focus on the various connections to cancer. This is a somewhat unique perspective and for that reason will be useful to the community. The authors might want to consider the following points prior to publication:

1. The logic for glutamine-derived aKG and NH₄ being important to contribute to ornithine is a bit circular. While the pathways described are correct, producing aKG from glutamine releases 2 NH₃ groups, although indeed additional NH₃ is ultimately

needed for arginine. To be more clear, the authors may want to revisit this discussion in the how arginine is produced section.

2. A role for arginine as an energy buffer is discussed in other organisms, including microbes. This is interesting, but the authors might also discuss that how different animals handle nitrogen is quite different, but many not secreting urea. Working this in might be helpful as this might not be evident to all readers.

3. Can the authors clarify what the evidence is for proline being rate limiting for structural protein synthesis? Certainly collagen contains a lot of proline, but I am not sure that statement is valid.

4. Adding redox cofactors to Figure 3 might help readers with the discussion of proline metabolism.

5. The authors mention clinical trials for depleting arginine. They might want to cite active trials, as the challenges they discussed with the trials have led (I think) to most abandoning the approach, although advocating for revisiting these approaches might be appropriate. However, the authors might offer some alternative selection strategies to do so given that ASS1 expression was not effective to select patients.

6. While a minor point, it might more correct to state phosphocreatine is important as an energy buffer rather than energy storage. I guess a role in energy storage is technically correct, the amount of energy stored is minimal.

Response to referees' suggestions

We thank the reviewers for their thoughtful input and careful reading of the review. We have made changes to address the reviewers' suggestions and we believe the changes make the manuscript more accessible and relevant to the broad readership of EMBO J.

Referee #1:

This is a well written and very timely review. The content is excellent and synthesizes the findings of a significant amount of literature into a balanced and comprehensive overview of our current understanding of arginine metabolism, especially in the setting of cancer. The paper also proposes important directions for the future to address current gaps in knowledge. I only have a few minor edits:

1) Urea Cycle loss was first documented in ccRCC in Ochocki et al. (2018) Cell Metabolism and should be cited as such.

The Ochocki et al. (2018) reference is now added at the start of the section for 'Endogenous arginine synthesis'.

2) The figures could be improved with the use of BioRender or a similar platform.

EMBO J plans to use their graphic artists to redraw the figures to help make the information more accessible. We appreciate the offer to improve the quality of the figures.

3) The sentence starting with "For example, in both HCC.." on page 21 has the two references reversed for the HCC and ccRCC findings.

We thank the reviewer for pointing this out. Both Missiaen et al. (2022) and Mossmann et al. (2023) used HCC as their primary tumor model. We have now modified the sentence as follows to include Ochocki et al. (2018) as the citation for ccRCC : "For example, in both ccRCC (Ochocki et al., 2018) and HCC (Missiaen et al., 2022; Mossmann et al, 2023) that express low levels of ASS1, the cancer cells concomitantly repress arginases (ARG1/2) and polyamine synthesis enzyme agmatinase (AGMAT) to maintain a high intracellular arginine concentration after assimilating arginine from the extracellular environment."

4) On page 22, the sentence "For example, TME in the murine..." appears to be grammatically incorrect and should be edited.

We have revised the sentence into: "For example, in the TME of murine PDAC tumors, arginine was

found to be present at 2–5 μM compared to 50-120 μM in serum (Apiz Saab et al., 2023; Lee et al., 2023; Sullivan et al, 2019).”

Referee #2:

Fung, Thompson and colleagues present an authoritative review on the biology of arginine in mammals, with a focus on the various connections to cancer. This is a somewhat unique perspective and for that reason will be useful to the community. The authors might want to consider the following points prior to publication:

1. The logic for glutamine-derived αKG and NH_4 being important to contribute to ornithine is a bit circular. While the pathways described are correct, producing αKG from glutamine releases 2 NH_3 groups, although indeed additional NH_3 is ultimately needed for arginine. To be more clear, the authors may want to revisit this discussion in the how arginine is produced section.

We have added more details on how mitochondrial glutamine-derived glutamate/ α -ketoglutarate and NH_4^+ can contribute to ornithine synthesis and serve as precursors for arginine synthesis under the subsection ‘Arginine can be produced from mitochondrial glutamine/glutamate’.

2. A role for arginine as an energy buffer is discussed in other organisms, including microbes. This is interesting, but the authors might also discuss that how different animals handle nitrogen is quite different, but many not secreting urea. Working this in might be helpful as this might not be evident to all readers.

Under the subsection ‘Urea and ammonia detoxification’, we have included a section about how different animal species detoxify ammonia, fishes can secrete ammonia directly into the aquatic environment while mammals secrete urea as by-product. Birds and reptiles on the other hand, employ the uric acid cycle.

3. Can the authors clarify what the evidence is for proline being rate limiting for structural protein synthesis? Certainly collagen contains a lot of proline, but I am not sure that statement is valid.

We agree with the reviewer that evidence for proline being rate limiting in structural protein synthesis is still indirect. Reports have shown that the addition of proline into proteins is slower compared to other amino acids (PMID: 27827794; PMID: 29100052). Furthermore, the consecutive addition of prolines can cause ribosomal stalling (PMID: 23239623) and hypusinated eIF5A elongation factor facilitates collagen synthesis and alleviates ribosomal stalling (PMID: 34447991). We have added these and additional details in the subsection ‘Proline’.

4. Adding redox cofactors to Figure 3 might help readers with the discussion of proline metabolism.

We felt adding the redox components in Figure 3 might be distracting to the main goal of the figure as reducing equivalents are used primarily to drive the ornithine conversion to proline among the different fates of arginine depicted in the figure.

Instead, under the subsection on 'Proline', we have detailed the redox requirements of proline metabolism, which requires mitochondrial NADH in the final step to produce proline from P5C through PYCR1/2. Mitochondrial P5C can be supplied from both arginine catabolism and glutamine-derived glutamate.

5. The authors mention clinical trials for depleting arginine. They might want to cite active trials, as the challenges they discussed with the trials have led (I think) to most abandoning the approach, although advocating for revisiting these approaches might be appropriate. However, the authors might offer some alternative selection strategies to do so given that ASS1 expression was not effective to select patients.

We agree that clinical trials for arginine depletion have largely resulted in a mixed response. Recently, a phase III clinical trial using ADI-PEG20 in HCC failed (PMID: 29659672). On the other hand, the combination of ADI-PEG20 together with other chemotherapies have led to improved outcomes for patients with pleural mesothelioma (PMID: 38358753).

For alternative approaches, we discussed how the overexpression of ASS and OTC urea cycle enzymes or amino acid transporters in CAR T cell therapy might be viable options from data in preclinical studies (PMID: 32573723; PMID: 36521029).

These new changes can be found in the section on 'Targeting arginine metabolism in cancer'.

6. While a minor point, it might more correct to state phosphocreatine is important as an energy buffer rather than energy storage. I guess a role in energy storage is technically correct, the amount of energy stored is minimal.

We have changed the wording from 'energy storage' in the abstract, Figure 2 and Figure 4 legends to 'energy buffering'.

Dear Craig,

Thank you for submitting the revised version of your review manuscript for consideration by the EMBO Journal.

I have carefully checked your amendments towards the experts' comments and found them to be well addressed and plausibly integrated into the revised text. I am thus very pleased to inform you that your review article has now been accepted for publication in the EMBO Journal.

I look forward to progressing swiftly towards online publication of this article!

Meanwhile
Best wishes to New York,

Daniel

Daniel Klimmeck, PhD
Senior Editor
The EMBO Journal
EMBO
Postfach 1022-40
Meyerhofstrasse 1

D-69117 Heidelberg
contact@embojournal.org
Submit at: <http://emboj.msubmit.net>